# Polyphenols Extracted from Shanxi-Aged Vinegar Inhibit Inflammation in LPS-Induced RAW264.7 Macrophages and ICR Mice via the Suppression of MAPK/NF-κB Pathway Activation

**DOI:** 10.3390/molecules26092745

**Published:** 2021-05-07

**Authors:** Peng Du, Jia Song, Huirui Qiu, Haorui Liu, Li Zhang, Junhan Zhou, Shengping Jiang, Jinyu Liu, Yu Zheng, Min Wang

**Affiliations:** 1Key Laboratory of Industrial Fermentation Microbiology (Tianjin University of Science and Technology), Ministry of Education, Tianjin 300457, China; dp9298@163.com (P.D.); 13199573023@139.com (H.Q.); 13205498828@163.com (H.L.); zl15774205290@163.com (L.Z.); zhoujh8877@163.com (J.Z.); 2State Key Laboratory of Food Nutrition and Safety, Tianjin Engineering Research Center of Microbial Metabolism and Fermentation Process Control, College of Biotechnology, Tianjin University of Science & Technology, Tianjin 300457, China; 3Research Center for Modern Analysis Techniques, Tianjin University of Science & Technology, Tianjin 300457, China; jiangshengping@tust.edu.cn (S.J.); liujy@tust.edu.cn (J.L.)

**Keywords:** Shanxi-aged vinegar, polyphenol extract, anti-inflammation, RAW264.7 macrophages, mice

## Abstract

Shanxi-aged vinegar, a traditional Chinese grain-fermented food that is rich in polyphenols, has been shown to have therapeutic effects on a variety of diseases. However, there has been no comprehensive evaluation of the anti-inflammatory activity of polyphenols extracted from Shanxi-aged vinegar (SAVEP) to date. The anti-inflammatory activities of SAVEP, both in RAW 264.7 macrophages and mice, were extensively investigated for the potential application of SAVEP as a novel anti-inflammatory agent. In order to confirm the notion that polyphenols could improve inflammatory symptoms, SAVEP was firstly detected by gas chromatography mass spectrometry (GC-MS). In total, 19 polyphenols were detected, including 12 phenolic acids. The study further investigated the protective effect of SAVEP on lipopolysaccharide-induced inflammation in RAW264.7 macrophages and ICR mice. The results showed that compared with those of the model group, SAVEP could remarkably recover the inflammation of macrophage RAW264.7 and ICR mice. SAVEP can normalise the expression of related proteins via the suppression of MAPK/NF-κB pathway activation, inhibiting the expression of iNOS and COX-2 proteins, and consequently the production of inflammatory factors, thus alleviating inflammatory stress. These results suggest that SAVEP may have a potential function against inflammation.

## 1. Introduction

Shanxi-aged vinegar (SAV) is a traditional Chinese fermented food that contains many active compounds [1]. The components, produced during the brewing of SAV, such as organic acids, sugar, peptides, amino acids, polyphenols, melanoidins, and flavour substances, all have health-related and therapeutic effects [2]. Among the functional molecules in SAV, polyphenol is an important antioxidant compound that has been identified. SAV contains a variety of polyphenols, including gallic acid, tannins, ferulic acid, catechin, quercetin, anthocyanins, p-coumaric acid, and resveratrol, which can prevent several diseases, such as cardiovascular diseases, hepatic damage, neurodegenerative diseases, cancer, and hyperlipemia [3,4]. Polyphenols have significant antioxidant effects, but their anti-inflammatory effects are also worthy of attention [5].

Inflammation is an important physiological response of the host against pathogenic infection or tissue damage, which plays a key role in infections [6], chronic diseases [7], cardiovascular diseases [8], and various human cancers [9,10]. Immune cells (such as macrophages) initiate an inflammatory response by increasing the production of nitric oxide (NO), reactive oxygen species (ROS), and inflammatory cytokines [11]. Low-level inflammation is necessary and beneficial to the human body [12], but excessive inflammation can lead to many diseases, such as asthma, rheumatoid arthritis, and atherosclerosis [13]. Drug therapy for inflammation is a traditionally effective method, but drug therapy often has side effects. Dexamethasone is an effective anti-inflammatory drug, but long-term use can lead to glucocorticoid withdrawal syndrome and digestive complications [14]. Therefore, anti-inflammatory ingredients derived from natural products are a good choice.

Lipopolysaccharide (LPS) is an effective stimulant for the release of pro-inflammatory cytokines and NO that induce vasodilation and increase vascular permeability, which can lead to cardiovascular dysfunction, lung function damage, acute renal insufficiency, septic shock, and even death [15]. LPS-induced cell damage and animal inflammation are common models of inflammation. The composition and mechanism of polyphenols in SAV are complex. The anti-inflammatory activity of polyphenols involves different targets: these targets can be subdivided into arachidonic acid-dependent pathways, such as cyclooxygenase (COX) inhibition, lipoxygenase inhibition, and phospholipase A2 inhibition. Within the arachidonic acid-independent pathways, nitric oxide synthase (NOS), nuclear factor κB (NF-κB), and NSAID activated gene-1 (NAG-1) are the targets of polyphenol. Moreover, polyphenols also affect the production of T helper type 1 and 2 cytokines [16,17]. 

In the arachidonic acid-dependent polyphenol pathway, part of its role is related to its antioxidant activity. This relationship has been studied for more than 20 years, and some papers presented data on the inhibition of COX-1 and COX-2 at transcriptional and enzymatic levels. However, according to recent studies, it is clear that polyphenols act on both pathways as antioxidants and modulators of gene expression. The type of activity can be distinguished. COX inhibition may explain the anti-inflammatory effect of reduced prostaglandin synthesis in the arachidonic pathway [17,18].

Epigallocatechin gallate (EGCG), a polyphenol in green tea, downregulates COX-2 in human mammary epithelial cells stimulated by 12-o-tetradecylphobos-13-acetate (TPA). It is suggested that the underlying mechanism is a decrease in the activation of extracellular signal-regulated protein kinase and p38 mitogen-activated protein kinases; upstream enzymes regulate the expression of COX-2. EGCG and green tea extract inhibit interleukin (IL)-1β-dependent inflammatory signal transduction and IL-8 gene expression via the NF-κB-dependent pathway. The authors suggest that the application of EGCG and its related compounds may represent a new pharmacological strategy for regulating the inflammatory response of the NF-κB pathway [19,20]. Different polyphenols affect different intracellular signalling pathways. In addition to the type of polyphenols, the disease and the stimulus applied are also important factors. Interactions and synergies between different polyphenols also affect different intracellular signalling pathways [21]. 

Even though polyphenols are beneficial to human health, the regulation mechanism of polyphenol extracts in vinegar in relation to inflammatory stress metabolism remains unknown. In order to further analyse the anti-inflammatory activity of SAVEP, firstly, polyphenols in SAVEP were isolated and quantitatively detected based on previous work; then, we further investigated the anti-inflammatory effects of SAVEP and the anti-inflammatory mechanism in LPS-induced RAW264.7 macrophages and ICR mice.

## 2. Results

### 2.1. Analysis of SAVEP

The determination of the total antioxidant activity of polyphenol extraction-A, polyphenol extraction-B, and polyphenol extraction-C was carried out using the DPPH, ABTS and FRAP methods, with ranges of 0.16 ± 0.02–1.36 ± 0.21 mmol TEAC/L, 1.61 ± 0.20–3.43 ± 0.13 mmol TEAC/L, and 0.22 ± 0.03–1.06 ± 0.06 mmol TEAC/L, respectively (Appendix A). The order of total antioxidant activity identified by the three methods is polyphenol extraction-A > polyphenol extraction-C > polyphenol extraction-B. The total antioxidant activity of the polyphenol extraction-A sample was the highest, and it was significantly different from the other samples (*p* < 0.05). It was shown that polyphenol extraction-A had strong free radical scavenging and reducing ability, and it exhibited the highest antioxidant activity among the three polyphenol extract samples. Our previous results suggest that antioxidant activity is positively correlated with anti-inflammatory effects [22]. Therefore, polyphenol extraction-A was selected as the most appropriate SAVEP in the subsequent research. 

Based on the GC-MS analysis and our previous work, a total intensity chromatogram image of SAVEP is shown in Figure 1. Referring to the quantitative method of GC-MS experiment conducted by Rabah [23], we chose the relative quantitative method to quantify the detected polyphenols. Our study showed a total of 703 peaks in the sample. Relative to those in the NIST 11 database, 19 polyphenols, including 12 phenolic acids, were found. Specific corresponding results are shown in Appendix A. The top five polyphenols by concentration are as follows: hydroferulic acid, vanillic acid, ferulic acid, 4-coumaric acid, and 4-hydroxybenzoic acid. The names of the five polyphenols corresponding to the polyphenol derivatives in Appendix A are ① benzenepropanoic acid, 3-methoxy-4 [(trimethylsilyl)oxy]-, trimethylsilyl ester; ② benzoic acid, 3-methoxy-4- [(trimethylsilyl)oxy]-, trimethylsilyl ester; ③ trimethylsilyl 3-methoxy-4-(trimethylsilyloxy)cinnamate, cinnamic acid; ④ p-(trimethylsiloxy)-, trimethylsilyl ester; and ⑤ benzoic acid, 4- [(trimethylsilyl)oxy]-, trimethylsilyl ester.

### 2.2. Effects of Different SAVEP Concentrations on Cell Viability and Dose Selection

Cell viability studies were conducted at five different SAVEP concentrations (100, 200, 400, 600, and 800 μg/mL). The maximum cytotoxic effect was observed at concentrations of 600 and 800 μg/mL, and the minimum was found at 400 μg/mL (Figure 2A). No significant difference was observed among concentrations of 100, 200, and 400 μg/mL or between RAW 264.7 macrophages and the control group at these three concentrations. Therefore, SAVEP concentrations in the range of 100–400 μg/mL had no effect on cell growth. 

As an important indicator of cellular inflammation, TNF-α can reflect the degree of inflammation. Figure 2B shows that SAVEP can effectively alleviate the inflammation induced by LPS, and each dose group had significant difference compared with the control group (*p* < 0.05). When the dose range was at 100–400 μg/mL, (concentration of TNF-α: 150.41 ± 5.95–135.19 ± 6.48 pg/mL), the expression of TNF-α quantity showed a declining trend. When the concentration was more than 600 μg/mL (concentration of TNF-α: 136.17 ± 7.05 pg/mL), the TNF-α content slightly increased because the increase in SAVEP concentration simultaneously protects and damages cells (Figure 2A). Therefore, 200 and 400 μg/mL were chosen as the low- and high-dose concentrations, respectively, for the next study.

### 2.3. Effects of SAVEP on LPS-Induced Inflammatory Cell Morphology 

The cell surface of the vehicle group was smooth without shrinkage, and that in the model group induced by LPS was morphologically heterogeneous with an uneven surface. The cells treated with SAVEP of different concentrations showed similar morphology to those of the vehicle group with a few pits and no atypia. These results show that SAVEP effectively alleviated the inflammatory stress caused by LPS and had a protective effect on cell morphology (Appendix A).

### 2.4. Effects of SAVEP on LPS-Induced Inflammatory Cell Nucleus Morphology

In addition to cell morphology, DNA damage is also an important indicator of cell health. LPS was used to induce inflammation on RAW264.7 cells, which were finally stained with DNA dye Hoechst 33,342 to reflect the degree of cell DNA damage. Figure 3 shows that under a fluorescence microscope, nuclear chromatin condensation or nuclear fragmentation gradually occurred in the RAW264.7 cells of the model group. Compared with the vehicle group (100 ± 6.28%), the value of blue fluorescence significantly increased by 254.24% in the model group (354.239 ± 36.57%) (*p* < 0.05). The nucleus morphologies in the SAVEP groups were significantly altered compared with that of the model group, and nucleus apoptosis was alleviated. The fluorescence intensity was reduced by 170.14% in the HD group (131.13 ± 18.86%) compared with the model group (*p* < 0.05). These results show that SAVEP at concentrations of 200 and 400 μg/mL could effectively protect the nucleus with inflammatory damage.

### 2.5. Effects of SAVEP on LPS-Induced Inflammatory Cell Mitochondrial Membrane Potential

The change in JC-1 from red fluorescence to green fluorescence can be used as an indicator of early apoptosis [24]. Figure 4A shows the fluorescence intensity observed under a fluorescence microscope. The green fluorescence intensity in the model group was increased compared with that in the vehicle group, whereas the red fluorescence intensity was decreased. The intracellular green fluorescence intensity was reduced in the SAVEP-treated groups compared with that in the model group, whereas the red fluorescence intensity was enhanced. The change in the HD group was more significant than that in the LD group. As shown in Figure 4B, the ratio of red/green fluorescence in the model group (52.53% ± 10.11%) was significantly reduced by 50.29% compared with that in the vehicle group (105.67% ± 12.23%) (*p* < 0.05), and the mitochondrial membrane potential was significantly reduced, resulting in severe cell apoptosis. The ratio of red/green fluorescence in the HD group (92.9% ± 13.31%) was increased by 76.85% compared with that in the model group (*p* < 0.05), indicating that HD could significantly increase mitochondrial membrane potential and rejuvenate injured cells. 

### 2.6. Effects of SAVEP on LPS-Induced Inflammatory Cell Apoptosis

Annexin V/PI staining was used to detect apoptosis. As shown in Figure 5, the apoptosis rate of LPS-induced cells significantly increased by 54.70 ± 6.80% (*p* < 0.05) compared with that of the vehicle group (37.48 ± 2.70%). However, compared with that of the model group, the apoptosis rate of SAVEP treatment gradually decreased with the increase in SAVEP concentration, thereby indicating a dose-dependent pattern. Notably, the SAVEP of 400 μL/mL had the best protective effect and reduced the apoptosis rate of damaged hepatocytes to 38.42 ± 1.50% (*p* < 0.05). Therefore, comprehensive apoptosis detection indicators showed that SAVEP could alleviate the inflammatory cell injury induced by LPS and that it has a protective effect on inflammatory cells.

### 2.7. Effects of SAVEP on LPS-Induced Inflammatory Cytokine Levels

IL-1β, IL-6, IL-18, MCP-1, and NO are typical inflammatory markers. Table 1 shows that all five inflammatory indicators in the model group increased to varying degrees and showed significant differences compared with those in the vehicle group, except IL-6 (*p* < 0.05). After SAVEP treatment, the inflammatory stress response of the cells was alleviated to a certain extent. The values of IL-1, IL-6, IL-18, and MCP-1 in the model group were 4.14 ± 0.64, 2.78 ± 0.20, 21.73 ± 0.60, and 3.06 ± 0.20 pg/mL, respectively, and that of NO was 5.30 × 10^−3^ ± 1.34 × 10^−4^ μmol/L. Compared with those in the model group, all indicators in the HD group showed a downward trend; the differences for IL-1β (2.53 ± 0.44 pg/mL), IL-6 (2.27 ± 0.26 pg/mL), and IL-18 (18.46 ± 1.21 pg/mL) were significant (*p* < 0.05), and the differences for IL-1β (2.85 ± 0.35 pg/mL) in the LD group with the model group were significant (*p* < 0.05). These results show that SAVEP exhibits a dose-dependent influence on relieving inflammatory stress response and that a high SAVEP concentration has a significant effect on repairing cellular inflammatory damage caused by LPS.

### 2.8. Effects of SAVEP on LPS-Induced Inflammatory Cell Inflammatory Protein Expression

P-p38, p38, JNK, P-JNK, ERK1/2, P-ERK1/2, iNOS, and COX-2 expression levels of RAW264.7 were analysed by Western blot analysis, and the results are shown in Figure 6. Relative to those in the vehicle group, the expression levels of iNOS and COX-2 of the model group were significantly increased by 94.92%, 607.14%, and 83.61%, respectively (all *p* < 0.05). The values of P-p38/p38 (37.84%), P-JNK/JNK (225.81%), and P-ERK1/2/ ERK1/2(151.77%) increased significantly in the model group compared with those in the vehicle group. This result indicates that LPS causes the inflammatory reaction and further increases the expression of inflammation-related proteins. Relative to those in the model group, the levels of P-p38/p38, P-JNK/JNK, and P-ERK1/2/ ERK1/2 were significantly decreased in the LD (65.67%, 54.46%, and 62.18%) and HD (67.65%, 59.41%, and 64.00%) groups (all *p* < 0.05), and iNOS was significantly decreased in the LD (42.42%) and HD (44.44%) groups (all *p* < 0.05). However, no significant change was observed in COX-2 expression in the LD and HD groups (*p* > 0.05). Thus, SAVEP supplementation could significantly downregulate the expression of iNOS, COX-2, p38, JNK, and ERK1/2 proteins.

### 2.9. Effects of SAVEP on Inflammatory Factor Levels in Mice

Changes in blood indexes are the most direct way to reflect the degree of inflammation in the body. Compared with those in the vehicle group, COX-2, CRP, NOS, IgM, IL-1β, IL-6, NO, and TNF-α index levels in the model group were increased by 72.99%, 28.25%, 43.21%, 39.41%, 101.42%, 11.56%, 100.45%, and 23.43% (all *p* < 0.05), respectively. However, IL-10, IgA, and IgG were decreased by 19.21%, 22.61% and 68.67% (all *p* < 0.05), respectively.

Compared with those in the model group, the indexes of COX-2, CRP, NOS, IgM, IL-1β, IL-6, NO, and TNF-α in the PC, LD, and HD groups presented a downward trend—the indexes in the HD group were increased by 38.41%, 22.48%, 22.49%, 19.60%, 45.43%, 12.04%, 47.62%, and 19.92% (*p* < 0.05), respectively. While the indexes of IL-10, IgA, and IgG presented an upward trend in the PC, LD, and HD groups, the indexes in the HD group were decreased by 14.41%, 25.11%, and 176.92% (*p* < 0.05), respectively (Figure 7). These results show that LPS-induced mice produced an inflammatory stress response, leading to a change in related inflammatory indicators. SAVEP intervention normalised the level of related inflammatory indicators, indicating that SAVEP can relieve the inflammatory stress response in mice and play a certain protective role in the body of mice. 

### 2.10. Effects of SAVEP on Inflammatory Protein Expression in LPS-Induced Inflammatory Mouse Liver

NF-κB, P-NF-κB, iNOS, and COX-2 levels in mouse liver were analysed by Western blot, and the results are shown in Figure 8. The P-NF-κB/NF-κB values of the model group significantly decreased by 41.49% compared with those of the vehicle group (all *p* < 0.05). The iNOS and COX-2 values of the model group significantly increased by 78.87% and 19.74%, respectively, compared with those of the vehicle group (all *p* < 0.05). The P-NF-κB/NF-κB values of the LD and HD groups were decreased by 22.21% and 24.38%, respectively (all *p* < 0.05). The values of iNOS (12.59% and 28.54%) and COX-2 (17.34% and 12.80%, respectively) were decreased in the LD and HD groups (all *p* < 0.05). SAVEP decreased the iNOS and COX-2 values compared with those in the model group. Therefore, high and low concentrations of SAVEP significantly inhibited the protein expression of the NF-κB pathway and downregulated the expression of iNOS and COX-2 proteins.

## 3. Discussion

Shanxi-aged vinegar is a famous traditional solid fermentation vinegar in China [1] and consists of organic acids, sugars, proteins, amino acids, polyphenols, melanoidins, and other nutrients and flavour components, all of which have health-related and therapeutic properties [2]. Even though polyphenols are important antioxidant and anti-inflammatory compounds widely found in vinegar [22], the regulation mechanism of polyphenol extracts in vinegar in relation to inflammatory stress metabolism remains unknown. In this study, types of polyphenols were detected by GC-MS; then, the anti-inflammatory effects of SAVEP and their potential mechanisms were investigated in LPS-induced RAW 264.7 macrophages and inflammatory mice.

Cell damage was mainly manifested as changes in cell morphology, DNA damage, and mitochondrial membrane potential imbalance, which could lead to apoptosis [25,26,27]. Normal cell morphology shows roundness and fullness without cell membrane rupture and depression. However, when inflammatory cell damage occurs, the corresponding cell morphology changes as reflected in cell shrinkage, anisotropy, and depression [15]. SAVEP blocks the morphologic change in LPS-stimulated macrophages to a certain extent and plays a protective role in inflammatory-damaged macrophages. Our findings are basically consistent with those regarding the recovery effect of *Rhododendron molle* leaf extract (Main component are polyphenols) on LPS-induced RAW264.7 cell morphology [24]. A certain concentration of LPS can induce DNA damage and trigger inflammatory response, eventually leading to apoptosis. The shrinkage, lysis, and production of apoptotic bodies are important markers of cell apoptosis [26]. The shrinkage of macrophage nucleus caused by LPS occurs because the increase in ROS in cells induced by LPS leads to the dysfunction of cell mitochondria and consequently to the shrinkage of the nucleus [28]. Our results showed that SAVEP could effectively restore LPS-induced nuclear injury of macrophages. Moreover, this phenomenon was dose dependent, and the recovery effect of a high SAVEP concentration on macrophages was significantly higher than that of a low concentration. In regard to the research of procyanidin A2 for the treatment of LPS-induced inflammatory macrophages, procyanidin A2 effectively improved nuclear contraction [29]. These results suggested that polyphenols play an important role in protecting the nucleus.

Protecting mitochondria from free radicals is an important aspect of antioxidant and anti-inflammatory reactions [30]. Imbalance of mitochondrial membrane potential is regarded as an important marker of apoptosis [31]. Our finding indicates the downward trend of the LPS-induced mitochondrial membrane potential of macrophages. The groups treated with SAVEP showed significantly reduced intracellular green fluorescence intensity and enhanced intracellular red fluorescence intensity. It is suggested that the mitochondrial membrane potential of macrophages was enhanced. The change in the HD group was more evident than that in the LD group, indicating that this influence is positively correlated with concentration. The destruction of mitochondrial transmembrane potential is often associated with apoptosis and is one of the earliest events during apoptosis [32]. Procyanidin A2 has a similar effect to that of SAVEP in restoring the mitochondrial membrane potential of macrophages [29]. LPS-induced macrophages were studied by flow cytometry to investigate the effect of SAVEP on apoptosis. Our results indicate that LPS induced irreversible damage to the macrophages and resulted in cell apoptosis, which may be related to nuclear damage and the imbalance of mitochondrial membrane potential. SAVEP can reduce the apoptosis rate of macrophages by 22.99% and 29.19%. These results show that SAVEP can help reduce the apoptosis of inflammatory macrophages. This trend was also confirmed by Qiu’s research on the polyphenol-rich extract from *Pleurotus eryngii*’s anti-inflammatory results for macrophages [33]. 

Pervasive cytokines (such as IL-6, IL-10, and TNF-α) play critical roles in activated macrophages [34]. Inflammation has a complex pathogenesis. Cells produce stress response after external stimulation, and the level of inflammatory factors increases. This increased value is a key factor to determine the inflammatory stress response of cells [15]. Some believe that the overexpression of inflammatory factors is an important cause of inflammation. Therefore, inhibiting a large number of pro-inflammatory cytokines is considered a potential therapeutic strategy to regulate inflammation-related diseases [35,36]. In our study, the values of TNF-α, IL-6, IL-18, IL-1β, MCP-1, and NO in the model group were all higher than those in the vehicle group. With SAVEP treatment, all of the indexes except NO showed a significant downward trend. The change in NO was small but still showed a downward trend. Excessive amounts of NO were associated with some inflammatory diseases [37]. Reducing NO, TNF-α, IL-6, and IL-1β overexpression helps reduce inflammation [15]. It has been reported that *Acalypha Australis* L. can alleviate the damage caused by inflammatory stress through reducing the expression of TNF-α and IL-1β in inflammatory macrophages [35]. Experiments were also conducted on mice to further investigate the anti-inflammatory effects of SAVEP and obtain additional results. The serum levels of IL-1β, IL-6, NO, and TNF-α in mice were consistent with those in inflammatory macrophages. Other inflammatory markers, such as IL-10, IgA, IgG, COX-2, and IgM, showed similar trends [38,39].

LPS triggers inflammation by binding to and activating receptor complexes on cell membranes. The receptor of LPS is the Toll-like receptor (TLR), which uses IL-1 signalling molecules (MyD88, IRAK, IRAK2, TRAF6, NIK, etc.) to mediate the activation of nuclear factor-κB (NF-κB) in cells induced by LPS [40]. In addition to NF-κB activation, TLR activates mitogen-activated protein kinases (MAPKs) [41]. NF-κB is one of the most important inducible transcription factors in mammals and plays a key role in their innate immune response and chronic inflammation [42]. This systemic inflammatory response induces the expression of proinflammatory mediators, namely, enzymes, cytokines, and chemokines [43]. LPS activates ERK1/2, JNK, and p38 MAPK, which ultimately control the activity of transcription factors regulating the expression of inflammation modulators, such as induced NO synthase (iNOS), cyclooxygenase-2 (COX-2), TNF-α, IL-1α, IL-1β, and IL-6 [44]. Inhibiting the activation of NF-κB and MAPK signalling pathways is an important treatment strategy for inflammatory diseases [15,45]. Many anti-inflammatory effects shown by polyphenols may be achieved by influencing the transcription network, regulating gene expression, changing the inflammatory cell cycle of recruitment, or homing to a certain extent, thereby eventually inhibiting the activation of NF-κB and MAPK signalling pathways [46].

The MAPK protein family, including JNK1/2, ERK1/2, and p38 MAPK, regulates the inflammatory response of cells to external factors. Studies have shown that polyphenols, such as kaempferol, chrysin, apigenin, and luteolin, have been shown to be active inhibitors of TNF-α stimulated airway epithelial cell ICAM-1 expression by inhibiting all three mitogen-activated protein kinases, ERK, JNK, and P38 [47]. Related proteins were examined to confirm the effect of SAVEP on MAPK (ERK1/2, JNK and P38) family protein expression of RAW264.7 cells. The evidence shows that the dietary polyphenol ferulaldehyde can inhibit LPS-induced phosphorylation of JNK, ERK1/2, and p38 MAPK proteins, reducing MAPK activation, inhibiting NF-kB activation, and effectively prolonging the lifespan of LPS-treated mice [48]. Our results show that SAVEP inhibits LPS-induced phosphorylation of ERK1/2, JNK, and P38 proteins. LPS-induced MAPK phosphorylation is involved in iNOS, COX-2 protein expression, and proinflammatory factor activation [47]. Therefore, the degree of inflammation is reduced under the action of SAVEP. Studies have shown that polyphenols inhibit the expression of COX-2 by inhibiting the expression of MAPKs. COX-2 is known as an inducible enzyme that produces, in most cases, large amounts of prostaglandins. The reduction of prostaglandins content reduces inflammation [21]. Recent studies have shown that in LPS-activated mouse macrophage RAW264, pro-delphinidin B-4 3′-O-gallate, and pro-delphinidin B2 3,3′ di-O-gallate inhibit COX-2 mRNA and protein expression in a dose-dependent manner [21,49,50]. SAVEP inhibits the expression of iNOS protein, which is related to the production of NO, NO is involved in the inflammatory process, and the reduction of NO content alleviates the inflammation. It has been shown that polyphenols inhibit NO release by inhibiting NOS enzyme expression and/or NOS activity [21]. In LPS-stimulated macrophages of J774 mice, hydroxytyrosol inhibits the expression of COX-2 and iNOS genes; these findings are consistent with our results.

Inhibition of NF-κB is generally recognised as an effective strategy for the treatment of inflammatory diseases [51]. The NF-κB pathway is an important and attractive therapeutic target for compounds that selectively interfere with NF-κB [21]. The NF-κB signalling pathway regulates inflammatory response. NF-κB is an eukaryotic transcription factor belonging to the family of NF-κB/Rel P50, which is a heterodimer and NF-κB P65 [52]. NF-κB expression and P-NF-κB maintain an appropriate dynamic equilibrium that prevents pathological inflammation from damaging the body, but this dynamic equilibrium also leads to pathological inflammatory damage and changes in inflammatory markers under pathological conditions [53]. These results suggest that high SAVEP concentrations significantly inhibit the abnormally high NF-κB expression and reduce inflammatory stress in the body. Polyphenols exert their anti-inflammatory activity by regulating NF-κB activation and participating in several steps of the activation process [21,54]. Studies have shown that epigallocatechin gallate (EGCG) markedly attenuated the myocardial injury after ischemia and reperfusion in rats, and this cardio-protection was associated with inhibition of NF-κB activation, which was consistent with our research [21]. COX-2 protein expression is not only related to MAPKS, but it is also closely related to NF-κB. SAVEP inhibited the activation of NF-κB, thereby affecting the expression of COX-2. The reduction of prostaglandin content associated with COX-2 protein was an important reason for the reduction of inflammation. Studies have found that the NF-κB signalling pathway is related to the expression of the iNOS protein. Preventing the binding of NF-κB to the iNOS gene promoter can cause inactivation of the iNOS protein and inhibit the transcription of iNOS. The result is that NO production decreases and inflammation is restored [55]. Tyrosol, along with lycopene and quercetin, inhibited the expression of COX-2 and iNOS genes in RAW 264.7 macrophages stimulated by gliadin in association with the NF-κB pathway. This finding is consistent with our findings [21]. The treatment of BV2 microglial cells with blueberry polyphenols has been shown to effectively reduce lipopolysaccharide (LPS)-induced pro-inflammatory mediators, such as nitric oxide (NO), interleukin 6 (IL-6), tumour necrosis factor-alpha (TNF-α), interleukin-1 beta (IL-1β), inducible NO, synthase (iNOS), and cyclooxygenase 2 (COX2). The effect was achieved by inhibiting the activation of the NF-κB signalling pathway and mediating the decrease in iNOS and COX2 protein expressions [56]. The inflammatory response is closely related to the immune response. Supplementation of taxifolin (a natural antioxidant polyphenol) significantly inhibited the NF-κB signalling pathway and significantly increased the secretions of IL-10, secretory immunoglobulin A, superoxide dismutase, and immunoglobulins (IgA, IgG, and IgM) in DSS-induced colitis mice [57]. Polyphenols isolated from virgin coconut oil attenuated cadmium-induced inflammation in rats. Inflammatory markers IL-6, CRP, and NO were significantly decreased [58]. The same trend was found in our study with the intervention of SAVEP; the NF-κB signalling pathway was inhibited, and the protein expressions of COX-2 and iNOS were decreased. The essence of inflammatory mediators is the expression of inflammatory proteins [56]. With the decrease in inflammatory protein expression, the contents of inflammatory mediators such as IL-6, CRP, NO, IL-1β, TNF-α, iNOS, and COX-2 decreased and IL-10, IgA, and IgG increased.

As a polyphenol extract from foods, SAVEP has shown good anti-inflammatory effects, which provides a new direction for the clinical treatment of inflammation. Although polyphenols have anti-inflammatory potential, they are commonly associated with low bioavailability and an extremely limited half-life, which leads to weak potency and poor pharmacokinetics [59]. We investigated the protective effect of SAVEP on LPS-induced RAW264.7 macrophages and ICR mice, using more cell lines and expanding the number of assays where necessary. Studies further exploring the pharmacodynamics and pharmacokinetics of SAVEP are warranted.

## 4. Materials and Methods

### 4.1. Materials and Chemicals

SAV samples with eight years of ageing time were obtained from Shanxi-aged vinegar Group Co. Ltd. (Shanxi, China). D101 macroporous adsorption resin was purchased from Shanghai McLean Biochemical Technology Co., Ltd. Methanol, ethyl acetate, normal hexane, and N-butyl alcohol (all AR grade) were obtained from Sigma-Aldrich (St. Louis, MO, USA). Bis(trimethylsilyl)trifluoroacetamide (BSTFA) + 1% trimethylchlorosilane (TMCS) and 2,4,5-trihydroxybenzoic acid (all chromatographic grade) were obtained from Sinopharm Chemical Reagent Co., Ltd. (Shanghai, China). Total antioxidant capacity assay kits with ABTS and FRAP were purchased from the Beyotime Institute of Biotechnology (Shanghai, China). DPPH was obtained from Sinopharm Chemical Reagent Co., Ltd. (Shanghai, China). 5,5′,6,6′-tetrachloro-1,1′,3,3′-tetraethylbenzimidazolyl-carbocyanine iodide (JC-1), cell counting kit-8 (CCK-8), annexin V/PI, alanine aminotransferase (ALT), aspartate aminotransferase (AST), monocyte chemoattractant protein-1 (MCP-1), nitric oxide synthase (NOS), nitric oxide (NO), C-reactive protein (CRP), immunoglobulin G (IgG), immunoglobulin A (IgA), immunoglobulin M (IgM), interleukin-1β (IL-1β), interleukin 6 (IL6), interleukin 10 (IL10), interleukin 18 (IL18), tumour necrosis factor α (TNF-α), and protein assay kits were obtained from Beijing Boxbio Science & Technology Co., Ltd. (Beijing, China). The primary antibodies against rabbit cyclooxygenase-2 (COX-2), inducible nitric oxide synthase (iNOS), Jun NH 2-terminal kinase (JNK), phospho-JNK (P-JNK), glyceraldehyde-3-phosphate dehydrogenase (GAPDH), p38 mitogen-activated protein kinase (p38 MAPK), phospho-p38 MAPK(P-p38 MAPK), extracellular signal regulated kinase(ERK1/2), phospho-ERK1/2(P-ERK1/2), and secondary horseradish peroxidase-labelled goat antirabbit antibodies were purchased from Abcam (Cambridge, UK). The antibodies against rabbit nuclear factor κB (NF-κB) and phosphorylated-NF-κB (P-NF-κB) were acquired from Cell Signaling Technology, Inc. (Beverly, MA, USA). Phosphate buffer saline (PBS), low melting agarose, sodium dodecyl sulphate, tris buffered saline tween (TBST), and lipopolysaccharides (LPS) were obtained from Beijing Solarbio Technology Co., Ltd. (Beijing, China). Dulbecco’s modified Eagle’s medium (DMEM) was obtained from Life Technologies (Grand Island, NY, USA). Other reagents were acquired from Sigma-Aldrich (Germany). Polyvinylidene fluoride (PVDF) membranes were obtained from Millipore (Schwallbach, Germany).

### 4.2. Polyphenol Extraction

Every 10 mL of SAV was filtered through an ultrafiltration membrane with a pore diameter of 10 kDa to remove the macromolecular impurities, such as proteins and melanoidins. The filtrate was then adsorbed to a D101 macroporous adsorption resin. After sample loading, the resin was washed with 10 times the volume of the resin of distilled water to remove saccharide and polar compounds, and it was then eluted with 10 times the volume of the resin of methanol. The preliminarily purified polyphenol extraction was obtained by vacuum evaporation of the methanol eluent. Components evaporated to dry were dissolved with 50 mL distilled water and then extracted with 50 mL ethyl acetate, n-hexane, and n-butanol in turn. The organic solvents were dried to obtain the polyphenol extracts; then, parts of the samples were dissolved with 10 mL distilled water and labelled as polyphenol extraction-A, polyphenol extraction-B, and polyphenol extraction-C. The other part was used for subsequent analysis.

Referring to our previous research methods [2], the total antioxidant activities of polyphenol extractions-A, -B and -C, which were redissolved with 10 mL distilled water, were compared by DPPH, ABTS, and FRAP methods. The group with the highest total antioxidant activity was chosen among the three groups, which was detected by GC-MS.

### 4.3. Analysis of Phenols of SAVEP

GC-MS has always been the preferred method for (unknown) material analysis due to its superior separation performance and high sensitivity. Mass spectral libraries are available to facilitate the identification of unknown substances and thus are suitable for the detection and analysis of polyphenols [60]. The polyphenols in the samples are non-volatile compounds, so chemical derivational steps are required to obtain volatile and thermally stable derivatives [22]. Based on our previous findings [22], we used 2,4,5-trihydroxybenzoic acid as an internal standard substance to perform relative quantitative detection of polyphenols in SAVEP. Before the experiment started, 0.003 g 2,4,5-trihydroxybenzoic acid was added to 10 mL of SAV according to the method in the Polyphenol Extraction Section, and then, the extraction of polyphenols was continued. In the final result, the ratio of the “peak area” of the internal standard to the polyphenol is the ratio of the concentration of the internal standard to the concentration of the polyphenol.

Derivative reaction conditions: The extract corresponding to 10 mL of vinegar was collected, and 1 mL of BSTFA + 1% TMCS was added. The reaction was conducted at 70 °C for 3 h in a water bath.

GC-MS detection conditions: The GC-MS system consisted of a gas chromatograph combined with a fast-scanning quadrupole mass spectrometer (GCMS QP2010, Shimadzu Corp, Kyoto, Japan) and an AOC-2000 autosampler. The injector temperature was 300 °C. Helium was used as carrier gas. The column was an HP-5 capillary column (30 m × 250 μm i.d. × 0.25 μm film thickness, Agilent Technologies, Cheadle, Cheshire, UK). The GC temperature ranging from 100 °C to 280 °C at 3 °C/min resulted in a total run time of 60 min. The transfer line was held at 300 °C. MS was operated at a maximum scan speed of 20,000 amu/s and at a scan range of 50–700 m/z, corresponding to a data acquisition rate of 33.3 s^−1^ (event time: 0.03 s). An ion source temperature of 200 °C was selected, and 70 eV EI spectra were recorded [22]. The National Institute Standard and Technology (NIST) was used to explain the GC-MS. The spectrum of unknown components was compared with the spectrum of the NIST (2011 version) library [61].

### 4.4. Cell Culture and Cell Viability Assay

RAW 264.7 macrophages were acquired from Wuhan Cell Bank, Chinese Academy of Sciences (Hubei, China) and cultured in DMEM medium containing 10% foetal bovine serum at 37 °C in a 5% CO_2_ incubator for 24 h. The cells were inoculated and treated with SAVEP of different concentrations for 24 h and then added with 1 μg/mL LPS for 12 h. The four groups were the vehicle group (vehicle, without LPS and SAVEP), model group (model, 1 μg/mL LPS), SAVEP high-dose group (HD, 1 μg/mL LPS and 400 μg/mL SAVEP), and SAVEP low-dose group (LD, 1 μg/mL LPS and 200 μg/mL SAVEP). Cell viability was determined by CCK-8 assay. RAW 264.7 macrophages (6 × 10^4^ cells/well) were inoculated in 96-well plates and incubated overnight prior to experimental intervention. The cells were then treated with samples of different concentrations for 24 h, added with 10 μL of CCK-8 in each coating, and incubated at 37 °C 5% CO_2_ for 2 h [62]. Optical density was then read at 450 nm using a microplate reader (Thermo Fisher Scientific, Multiskan Sky, Waltham, MA, USA). Absorbance was compared between the sample and vehicle group.

### 4.5. Scanning Electron Microscopy (SEM)

RAW264.7 macrophages (1 × 10^5^ cells/well) were cultured into six-well plates per hole at 37 °C and 5% CO_2_ for 24 h. Pre-cooled PBS buffer was added to the hole containing the glass sheet for cleaning twice. Afterwards, 2.5% glutaraldehyde fixative was added to the hole and placed in the refrigerator at 4 °C for 3 h. The samples were washed again with PBS buffer. The water in the sample was removed through gradient dehydration with ethanol. The concentration of dehydrating agent was 30%, 50%, 70%, 80%, 90%, and 100% (once), and the dehydration time was 15 min per step. The samples were dried in a critical point dryer for 2 h. SEM was conducted at 10 kV. The samples were sprinkled onto conductive glue on an aluminium SEM stub and sputter coated with gold.

### 4.6. Hoechst 33342/Propidium Iodide (PI) Staining

RAW264.7 macrophages (1 × 10^5^ cells/well) were cultured into a six-well plate per hole at 37 °C and 5% CO_2_. After 6 h, the cells were washed with PBS buffer three times. After the final wash, DMEM cell culture medium was added. The cells were then stimulated by LPS for 24 h, and some were added with SAVEP. Briefly, 1 μL of Hoechst 33,342 dye (1 mg/mL) was added to each well. The cells were incubated in darkness at 37 °C for 10 min and then washed twice with pre-cooled PBS buffer [62,63]. Red and green fluorescence spots were observed by a fluorescence microscopy (Thermo Fisher Scientific, EVOS M5000, Waltham, Massachusetts, USA) equipped with a DXM1200C colour digital camera (Nikon Corporation, Kumagaya, Saitama, Japan) and processed using NIS-Elements Basic Research v2.2 software.

### 4.7. JC-1 Staining

RAW 264.7 macrophages (1 × 10^5^ cells/well) were cultured into a six-well plate per hole at 37 °C and 5% CO_2_. After 6 h, the cells were washed with PBS buffer three times. After the final wash, DMEM cell culture medium was added. The cells were then stimulated by LPS for 24 h, and some were added with SAVEP. Concentrated JC-1 solution was diluted into a 5 g/mL working solution using DMEM medium. After mixing, the concentrated solution was added to the pore plate containing cells. The solution was placed in an incubator and incubated in darkness for 20 min [64]. Red and green fluorescence spots were observed by fluorescence microscopy (Thermo Fisher Scientific, EVOS M5000, Waltham, MA, USA).

### 4.8. Annexin V-FITC/PI Staining

RAW264.7 macrophages (1 × 10^5^ cells/well) were cultured into a six-well plate per hole at 37 °C and 5% CO_2_. After 6 h, the cells were washed with PBS buffer three times, collected, centrifuged at 4000 r/min for 5 min, and resuspended with cold PBS buffer.

The samples were tested by flow cytometry (Thermo Fisher Scientific, Attune NxT, Waltham, MA, USA). FlowJo Collectors Edition (Tree Star, Ashland, OR, USA) was used as acquisition software. The samples were diluted with 1 × PBS to produce solutions in which events were detected at a maximum rate of 1000 MPs per second. The dilutions ranged from 1:5 to 1:50 depending on the sample concentration. With predetermined parameters, events that exceed the minimum size threshold were collected. Data were collected until 10,000 events had been counted or 1 min had elapsed after the count, subtracting background events for buffering only, and the counts/μL sample was calculated using the recorded flow rate of the machine. 

Samples were stained separately with annexin V/PI binding solution (195 μL), annexin V dye (10 μL), and propidium iodide (5 μL PI), and they were incubated in dark for 20 min. A total of 500 μL was prepared for each sample [65]. The data were analysed using FlowJo analysis software.

### 4.9. Analysis of Cytokine Expression and Mouse Serum Index

RAW 264.7 macrophages were pre-treated with LPS (1 μg/mL) in six-well plates for 24 h and cultured with different samples for another 24 h. TNF-α, IL-1β, IL-6, IL-18, MCP-1, and NO levels in the supernatants were measured by using commercial kits in accordance with the manufacturer’s standards and protocols [63].

Mouse serum supernatants were collected through centrifugation at 5000 r/min for 15 min. Relevant indicators were analysed with commercial kits in accordance with the manufacturer’s standards and protocols [22].

### 4.10. Western Blot Analysis

Methods of protein extraction and incubation from RAW264.7 macrophages and mouse liver tissue were performed as described [62,63,66]. In brief, RAW264.7 macrophages or the liver tissues of mice were lysed and centrifuged at 12,000 rpm for 30 min to remove the fragments and impurities. The proteins were resolved by 12% SDS-polyacrylamide jell electrophoresis. Proteins were transferred to PVDF membranes and incubated for 2 h with skim milk in TBST. Blots were probed using the following primary antibodies: anti-p38 antibody (1:1000), anti-P-p38 antibody (1:1000), anti-ERK1/2 antibody (1:1000), anti-P-ERK1/2 antibody (1:1000), anti-NF-κB antibody (1:1000), anti-P-NF-κB antibody (1:1000), anti-iNOS antibody (1:1000), anti-COX-2 antibody (1:1000), anti-P-JNK antibody (1:1000), anti-JNK antibody (1:1000), and anti-GADPH antibody (1:1000). After the specimens were incubated with the primary antibody, the secondary antibodies were added. Western blot bands were quantified using Image J.

### 4.11. Animals, Fodder and Treatment

All experiments involving mice were conducted in accordance with the protocol approved by the animal protection and use committee of Tianjin University of Science & Technology. Male ICR mice (18–22 g) were purchased from the China National Institute for Food and Drug Control (License No.: SCXK (Beijing) 2016-002) and raised in the animal experiment centre (SPF) of the College of Biological Engineering, Tianjin University of Science and Technology. The indoor ventilation conditions were good, the relative humidity was controlled at 55% ± 5%, and the temperature was 23 ± 2 °C. Conventional diet and free intake of food and water were administered for 1 week for the subjects to adapt to the environment. A regular 12 h day/12 h night cycle was adopted [22]. 

The conventional feed components contained 59% carbohydrates, 21.1% proteins, 4.9% fibres, 4.2% fat, 8% ash, 1% phosphorus, and 1.8% calcium. The conventional feed was purchased from Beijing Keao Xieli Feed Co., Ltd.

The mice were randomly divided into five groups (*n* = 6), namely, vehicle, model (LPS-induced), PC (positive control, LPS-induced, and 5 mg/kg dexamethasone) and test groups (LD/HD), which received different doses of SAVEP (LPS-induced and 0.625/1.250 mL/kg SAVEP). All groups were intraperitoneally injected with 1.5 mg·kg^−1^ LPS in the first day, except for the vehicle group that was intraperitoneally injected with the same amount of normal saline. The experimental group was intraperitoneally gavaged with different doses of SAVEP for 7 days after continuous intervention, and the model and experimental groups were intraperitoneally injected with 1.5 mg·kg^−1^ LPS again 1 h after intervention on day 7. After 30 min, the mice were euthanised by an overdose of anaesthesia, and blood was collected. Finally, the liver was collected and placed in a −80 °C environment.

All animal procedures were performed in accordance with the Guidelines for Care and Use of Laboratory Animals (Ministry of Science and Technology of China, 2006) and were supported by the Institutional Animal Committee of Tianjin University of Science & Technology (TUST20180522).

### 4.12. Statistical Analysis

Each sampling dataset was repeated at least three times. Statistical analysis was performed using a one-way ANOVA with Tukey’s test. Origin 9.1 was used to draw the figures, and experimental data are expressed as mean ± SD with *p* < 0.05 as the standard for significant difference.

## 5. Conclusions

In summary, we detected 19 types of polyphenols in SAVEP. SAVEP can significantly improve LPS-induced inflammation, restore cell morphology, repair damaged nuclei, and recover the abnormal mitochondrial membrane potential, thus reducing cell apoptosis. SAVEP can inhibit the activation of the MAPK pathway in RAW264.7 and the NF-κB pathway in mice, inhibiting the expression of iNOS and COX-2 proteins and consequently the production of inflammatory factors, thus alleviating inflammatory stress. This compound can normalise the expression of related proteins and consequently the production of inflammatory factors, thus alleviating inflammatory stress. Therefore, our results suggest that SAVEP may be a novel strategy against inflammation. 

## Figures and Tables

**Figure 1 molecules-26-02745-f001:**
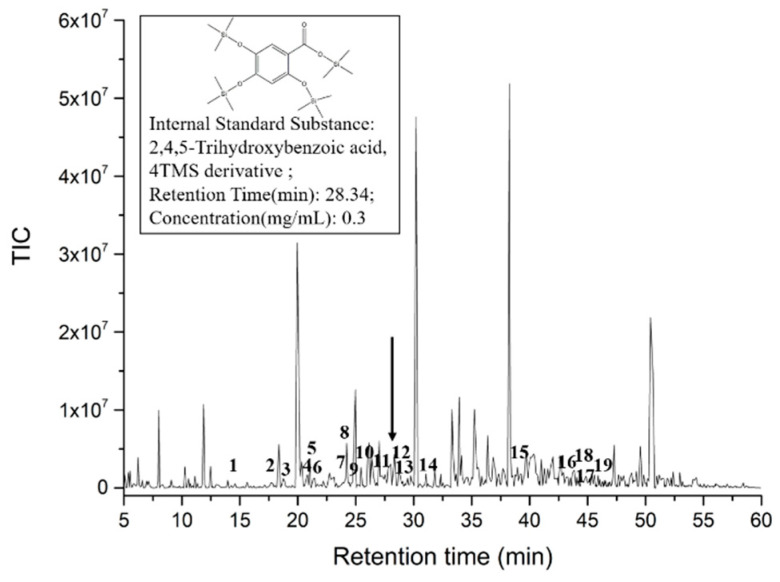
Total intensity chromatogram (TIC) images of SAVEP.

**Figure 2 molecules-26-02745-f002:**
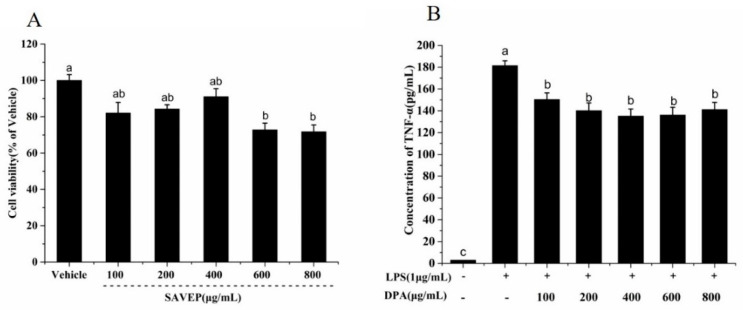
Screening of anti-inflammatory activity of the SAVEP. (**A**) Effects of different concentrations of SAVEP on cell viability. (**B**) Screening of anti-inflammatory activity concentration of SAVEP. Statistical analysis was performed using a one-way ANOVA with Tukey’s test. Data represent the mean ± SD (*n* = 6). Different letters represent significant differences between groups (*p* < 0.05).

**Figure 3 molecules-26-02745-f003:**
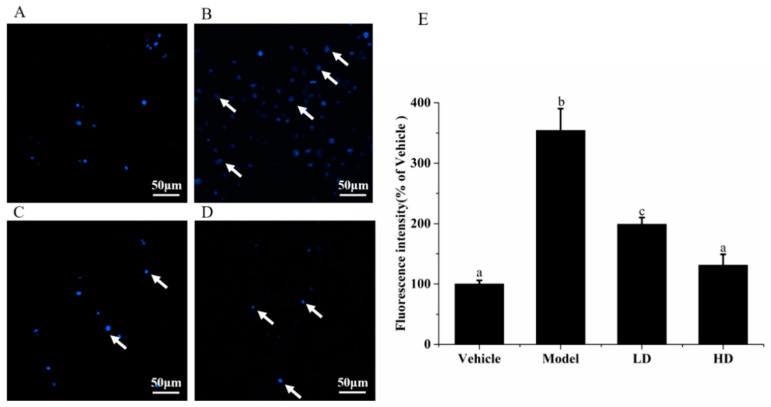
Effects of SAVEP on RAW264.7 cell nuclear morphology of LPS-induced inflammatory injury with fluorescence electron microscope. (**A**) Vehicle group. (**B**) Model group. (**C**) LD group (**D**) HD group. (**E**) Fluorescence intensity (of vehicle). Statistical analysis was performed using a one-way ANOVA with Tukey’s test. Data represent the mean ± SD (*n* = 6). Different letters represent significant differences between groups (*p* < 0.05).

**Figure 4 molecules-26-02745-f004:**
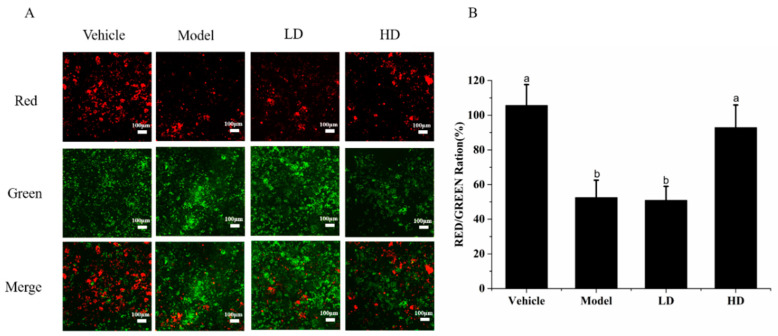
Effects of SAVEP on RAW264.7 cell mitochondrial membrane potential (MMP) of LPS-induced inflammatory injury. (**A**) Fluorescence intensity was observed by fluorescence microscope. (**B**) Fluorescence intensity of red/green ratio. Statistical analysis was performed using a one-way ANOVA with Tukey’s test. Data represent the mean ± SD (*n* = 6). Different letters represent significant differences between groups (*p* < 0.05). A larger version of (**A**) is shown in Appendix A.

**Figure 5 molecules-26-02745-f005:**
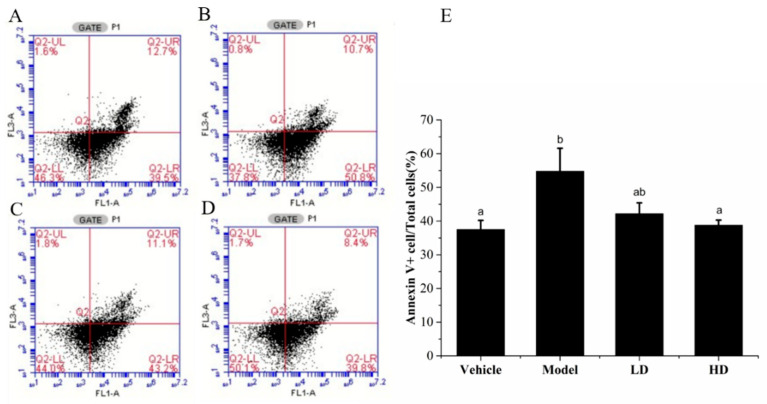
RAW264.7 cells were stained with annexin V-FITC and propidium iodide (PI) as detected by flow cytometer. (**A**) Vehicle group. (**B**) Model group. (**C**) LD group. (**D**) HD group. Statistical analysis was performed using a one-way ANOVA with Tukey’s test. Data represent the mean ± SD (*n* = 6). Different letters represent significant differences between groups (*p* < 0.05).

**Figure 6 molecules-26-02745-f006:**
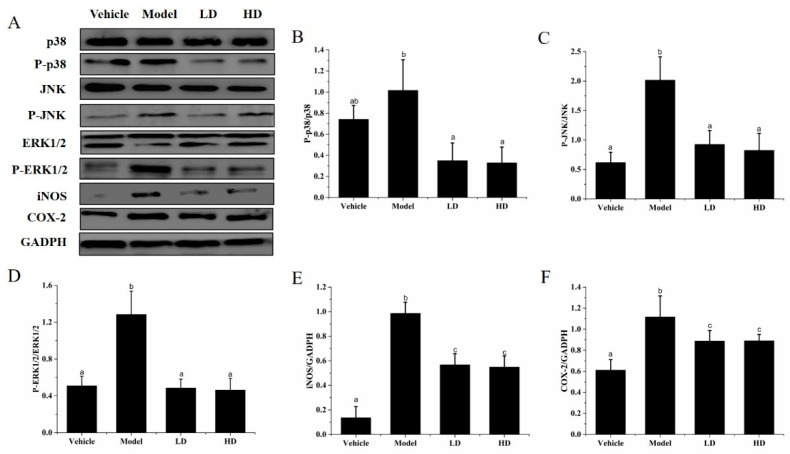
Effect of SAVEP on inflammatory protein expression in inflammatory-damaged RAW264.7 cells. (**A**) Expression levels of proteins were detected through Western blot analysis. (**B**) Quantitative analysis of P-p38/p38 protein level. (**C**) Quantitative analysis of P-JNK/JNK protein level. (**D**) Quantitative analysis of P-ERK1/2/ERK1/2 protein level. (**E**) Quantification of iNOS protein expression. (**F**) Quantification of the COX-2 protein expression. Statistical analysis was performed using a one-way ANOVA with Tukey’s test. Data represent the mean ± SD (*n* = 6). Different letters represent significant differences between groups (*p* < 0.05).

**Figure 7 molecules-26-02745-f007:**
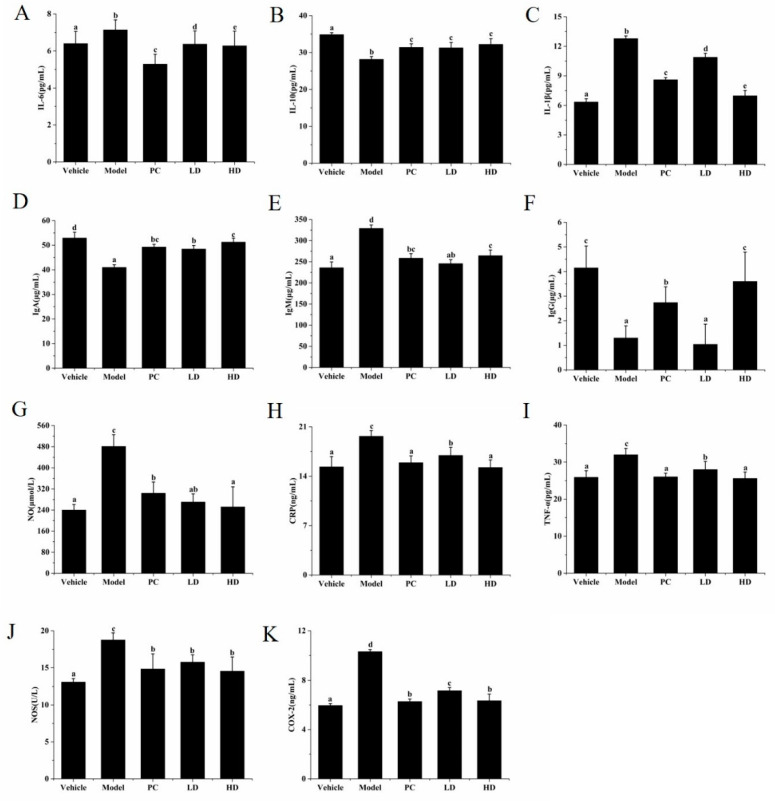
Effects of SAVEP on serum indexes in mice. Quantitative analysis of IL-6 (**A**), IL-10 (**B**), IL-1β (**C**), IgA (**D**), IgM (**E**), IgG (**F**), NO (**G**), CRP (**H**), TNF-α (**I**), NOS (**J**), and COX-2 (**K**). Statistical analysis was performed using a one-way ANOVA with Tukey’s test. Data represent the mean ± SD (*n* = 6). Different letters represent significant differences between groups (*p* < 0.05).

**Figure 8 molecules-26-02745-f008:**
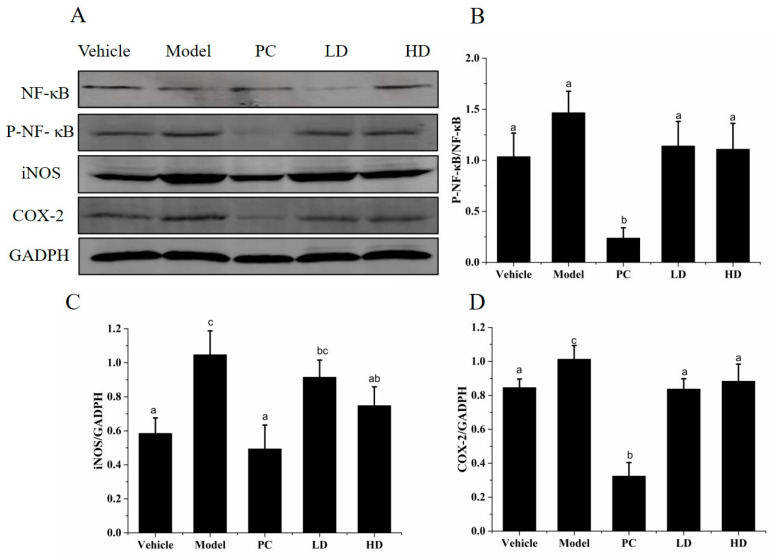
Effects of SAVEP on inflammatory protein expression in LPS-induced mice. (**A**) Expression levels of proteins were detected through Western blot analysis. (**B**) Quantitative analysis of P-NF-κB/ NF-κB protein level. (**C**) Quantitative analysis of iNOS protein level. (**D**) Quantification of COX-2 protein expression. Statistical analysis was performed using a one-way ANOVA with Tukey’s test. Data represent the mean ± SD (*n* = 6). Different letters represent significant differences between groups (*p* < 0.05).

**Table 1 molecules-26-02745-t001:** Effects of SAVEP on inflammatory cytokines levels.

	Vehicle	Model	LD	HD	*p*-Value
IL-1β(pg/mL)	2.31 ± 0.20 ^a^	4.14 ± 0.64 ^b^	2.85 ± 0.35 ^a^	2.53 ± 0.44 ^a^	0.004
IL-6(pg/mL)	2.46 ± 0.13 ^ab^	2.78 ± 0.20 ^b^	2.51 ± 0.11 ^ab^	2.26 ± 0.26 ^a^	0.049
IL-18(pg/mL)	17.80 ± 0.51 ^a^	21.73 ± 0.60 ^b^	20.41 ± 0.49 ^b^	18.46 ± 1.21 ^a^	0.008
MCP-1(pg/mL)	2.21 ± 0.20 ^a^	3.06 ± 0.20 ^b^	2.87 ± 0.35 ^b^	2.75 ± 0.25 ^ab^	0.019
NO(μmol/L)	4.60 × 10^−3^ ± 2.00 × 10^−4 a^	5.30 × 10^−3^ ± 1.34 × 10^−4 b^	5.39 × 10^−3^ ± 5.06 × 10^−4 b^	5.20 × 10^−3^ ± 1.38 × 10^−4 ab^	0.033

Statistical analysis was performed using a one-way ANOVA with Tukey’s test. Data are presented as mean ± SD (*n* = 6). Values in the same row with different letters are significantly different (*p* < 0.05).

## Data Availability

All generated and analysed data used to support the findings of this study are included within the article.

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
