# Peer review of "Polyphenols Extracted from Shanxi-Aged Vinegar Inhibit Inflammation in LPS-Induced RAW264.7 Macrophages and ICR Mice via the Suppression of MAPK/NF-κB Pathway Activation"

_molecules, 2021, doi:10.3390/molecules26092745_

Round 1

Reviewer 1 Report

“Polyphenols extracted from Shanxi-aged vinegar inhibits inflammation in LPS-induced RAW264.7 macrophages and ICR mice via the suppression of MAPKs/NF-kB pathway activation” is an interesting manuscript that investigates the anti-inflammatory properties of polyphenols contained in Shanxi Aged Vinegar (SAVEP). The authors focused on MAPK pathways and investigated LPS-induced inflammation in vitro and in vivo. The level of English is appropriate and the discussion well written. Few comments:

Major comment:

The result section is too verbose (especially sections 2.1 to 2.5). Authors should synthesize each section to a clear description of the results. The significance of the performed analysis and their meaning with the current literature should be moved to the discussion.

Authors discuss the relevance of LPS-induced ERK 1/2 activation, but no data is present. Authors should analyze the phosphorylation of ERK 1/2  as it would widen the significance of SAVEP anti-inflammatory properties on another member of the MAPK family and strengthen their discussion.

Limitations of the study and their significance to their possible clinical intervention should be included in the discussion as the bioavailability and half-life of Polyphenols is extremely limited. Authors are encouraged to analyze the plasmatic levels of SAVEP in their animal model to better discuss the relevance of their results.

The manuscript contains some clerical errors, a more throughout review is encouraged.

Minor comments:

Table 2 is difficult to read. The manuscript would benefit from moving data to a bar graph.

The result section should be written with normal font and not italic. (italic)

The type of ANOVA performed should be described in each figure legend and stars should be added to each of the figures.

Section 2.7 Data in the text and in Table 1 can be enlisted with just one or 2 decimal numbers (e.g. 4.14, 2.77, 21.79, ..). In table 1 an additional column with the p value should be added.

Reviewer 2 Report

Dear Authors,

The manuscript describe the first time for a detailed investigation of anti-inflammatory potential from Shanxi-aged vinegar purchased at Shanxi-aged vinegar Group Co. Ltd. (China). This sample was characterized by GC-MS, followed by intensive assays were carried out to evaluate anti-inflammatory activity from this sample, which including tracking the expression of iNOS and COX-2 and pro-inflammatory proteins. This finally allowed Authors to determine its anti-inflammatory mechanism pathway.

1) The whole result's section, from 2.1 to 2.10,  an example, page 3, lines at 110-114, "GC-MS has always been the preferred method for unknown material analysis due to superior separation ...", these sentences are not necessary as it is very common that we used GCMS or LCMS to profile chemical constituents of a crude extract. Another example, page 4, lines 153-154, "SEM can observe cell morphology...". More example, page 4, line 168, "The higher the blue fluorescence value, the greater the nuclear damage". Again, page 5, lines 183-184, "Imbalance of mithochondrial membrane potential is regarded as an important marker of apoptosis.". Page 7, lines 239-241, "LPS-induced inflammatory stress firstly up-regulates anti-inflammatory-related genes and proteins in the body, which in turn upregulate the expression of inflammatory-related signal pathways when the regulation ability exceeds the body’s own anti-inflammatory ability".

Many of these fundamental statements have no citation, and not necessary to be included in research article. These sentences looked like from thesis/dissertation.

2) Page 3, lines at 110-114, GC-MS library and derivatized method should mention in detailed and move from "results" to "methodology".

3) From page 13, lines 463-464, the methanolic crude extract was partitioned between water and series of organic solvents such as EtOAc, n-hexane, and n-BuOH, subsequently each fraction was dried and re-dissolved in water for further assay. The organic fractions can be dissolved in water? 

4) From materials and methods page, the methodology for cell culture, Hoechst 33342 and propidium iodide staining, Western-blot, cytotoxic and anti-inflammatory assays were not cited. Authors may consider below MDPI references.

a) Bioactive Cembranoids from the Soft Coral Genus Sinularia sp. in Borneo.  Mar. Drugs 2018, 16, 99; DOI: 10.3390/md16040099

b) Fucoidan Purified from Sargassum polycystum Induces Apoptosis through Mitochondria-Mediated Pathway in HL-60 and MCF-7 Cells.  Mar. Drugs 2020, 18, 196; DOI: 10.3390/md18040196

5) The microscope of Figure 4 is too small, and not clear.

6) The camera/digital capture for microscopic image should be mention. 

7) Concentration and settings on the flow cytometer should be mention.

8) Western blotting section, the manufacturer of chemical reagents such as the antibodies, incubation and centrifugation should be detailed out or with citation.

9) Conventional diet for mice should be detailed out by providing a longer information.

10) Using more cell lines and expanding the number of assays would establish a firmer basis for claiming anti-inflammatory activity.

Round 2

Reviewer 1 Report

The authors have followed reviewer's comments and the manuscript has been updated. 

The new Figure 7 is difficult to read. Authors should add the type of cytokine in the y axis.

Reviewer 2 Report

Dear Authors,

Thank you for addressing the comments positively.

Author Response

Thank you very much for your feedback regarding our manuscript # molecules-1196806 submitted to Molecules. We greatly appreciate your thoughtful comments.